# Pericentromeric hypomethylation elicits an interferon response in an animal model of ICF syndrome

Srivarsha Rajshekar[1,2,3], Jun Yao[2], Paige K Arnold[4], Sara G Payne[5], Yinwen Zhang[3], Teresa V Bowman[5], Robert J Schmitz[6], John R Edwards[7], Mary Goll[2,6]*

[1]Program in Biochemistry and Structural Biology, Cell and Developmental Biology, and Molecular Biology, Weill Cornell Graduate School of Medical Sciences, Cornell University, New York, United States; [2]Developmental Biology Program, Memorial Sloan Kettering Cancer Center, New York, United States; [3]Institute of Bioinformatics, University of Georgia, Athens, United States; [4]Louis V. Gerstner Jr. Graduate School of Biomedical Sciences, Memorial Sloan Kettering Cancer Center, New York, United States; [5]Department of Developmental and Molecular Biology, Albert Einstein College of Medicine, New York, United States; [6]Department of Genetics, University of Georgia, Georgia, United States; [7]Department of Medicine, Center for Pharmacogenomics, Washington University in St. Louis School of Medicine, Missouri, United States

*For correspondence:
Mary.Goll@uga.edu

**Competing interests:** The authors declare that no competing interests exist.

**Abstract** Pericentromeric satellite repeats are enriched in 5-methylcytosine (5mC). Loss of 5mC at these sequences is common in cancer and is a hallmark of Immunodeficiency, Centromere and Facial abnormalities (ICF) syndrome. While the general importance of 5mC is well-established, the specific functions of 5mC at pericentromeres are less clear. To address this deficiency, we generated a viable animal model of pericentromeric hypomethylation through mutation of the ICF-gene *ZBTB24*. Deletion of zebrafish *zbtb24* caused a progressive loss of 5mC at pericentromeres and ICF-like phenotypes. Hypomethylation of these repeats triggered derepression of pericentromeric transcripts and activation of an interferon-based innate immune response. Injection of pericentromeric RNA is sufficient to elicit this response in wild-type embryos, and mutation of the MDA5-MAVS dsRNA-sensing machinery blocks the response in mutants. These findings identify activation of the innate immune system as an early consequence of pericentromeric hypomethylation, implicating derepression of pericentromeric transcripts as a trigger of autoimmunity.

**Editorial note:** This article has been through an editorial process in which the authors decide how to respond to the issues raised during peer review. The Reviewing Editor's assessment is that all the issues have been addressed (see decision letter).
DOI: https://doi.org/10.7554/eLife.39658.001

## Introduction

In vertebrate genomes, the majority of cytosine residues within CpG dinucleotides are methylated at the 5 position of the cytosine ring (5-methylcytosine, 5mC) (*Suzuki and Bird, 2008*). 5mC is established by the de novo DNA methyltransferases of the Dnmt3 family, and is propagated by the maintenance DNA methyltransferase, Dnmt1 (*Goll and Bestor, 2005*). In mice, frogs and zebrafish, mutation or morpholino-mediated depletion of *Dnmt1* results in extensive genome-wide methylation

**eLife digest** Cells package DNA into structures called chromosomes. When cells divide, each chromosome duplicates, and a structure called a centromere initially holds the copies together. The sequences of DNA on either side of the centromeres are often highly repetitive. In backboned animals, this DNA normally also has extra chemical modifications called methyl groups attached to it. The role that these methyl groups play in this region is not known, although in other DNA regions they often stop the DNA being 'transcribed' into molecules of RNA.

The cells of people who have a rare human genetic disorder called ICF syndrome, lack the methyl groups near the centromere. The methyl groups may also be lost in old and cancerous cells.

Researchers often use 'model' animals to investigate the effects of DNA modifications. But, until now, there were no animal models that lose methyl groups from the DNA around centromeres in the same way as seen in ICF syndrome.

Rajshekar et al. have developed a new zebrafish model for ICF syndrome that loses the methyl groups around its centromeres over time. Studying the cells of these zebrafish showed that when the methyl groups are missing, the cell starts to transcribe the DNA sequences around the centromeres. The resulting RNA molecules appear to be mistaken by the cell for viral RNA. They activate immune sensors that normally detect RNA viruses, which triggers an immune response.

The new zebrafish model can now be used in further studies to help researchers to understand the key features of ICF syndrome. Future work could also investigate whether the loss of methyl groups around the centromeres plays a role in other diseases where the immune system attacks healthy tissues.

DOI: https://doi.org/10.7554/eLife.39658.002

loss and embryonic lethality (*Anderson et al., 2009*; *Lei et al., 1996*; *Rai et al., 2006*; *Stancheva and Meehan, 2000*). In these species, global methylation deficiencies are linked to a variety of adverse outcomes including deregulation of gene expression, derepression of transposons, elevated levels of DNA damage and increased genome instability during mitosis (*Smith and Meissner, 2013*). Recent studies have further linked global hypomethylation to activation of antiviral signaling pathways in zebrafish mutated for *dnmt1* and in cancer cells treated with the DNA methyltransferase inhibitor 5-azacytidine (*Chernyavskaya et al., 2017*; *Chiappinelli et al., 2015*; *Roulois et al., 2015*). While these studies reinforce the general importance of DNA methylation in vertebrate development and tissue homeostasis, the extensive genome-wide loss of methylation in these models makes it difficult to assign significance to methylation deficiencies at any particular subclass of sequence.

The pericentromeric satellite sequences that juxtapose chromosome centromeres represent an essential structural component of chromosomes and a significant source of 5mC in vertebrate genomes. These highly repetitive sequences appear particularly susceptible to methylation loss in cancer and senescent cells, although the consequences of this hypomethylation are not well understood (*Enukashvily et al., 2007*; *Fanelli et al., 2008*; *Nakagawa et al., 2005*; *Narayan et al., 1998*; *Qu et al., 1999*; *Suzuki et al., 2002*; *Tsuda et al., 2002*). In contrast to global hypomethylation, loss of 5mC at pericentromeric repeats is compatible with human development. Individuals with the rare, autosomal recessive disorder Immunodeficiency, Centromere and Facial anomalies (ICF) syndrome show extensive hypomethylation of pericentromeric repeats, while methylation across the rest of the genome is relatively intact (*Tuck-Muller et al., 2000*; *Velasco et al., 2018*; *Weisenberger et al., 2005*). Affected individuals usually die in late childhood or early adulthood, and exhibit variable symptoms including immunoglobulin deficiency, facial dysmorphism, growth retardation and a generalized failure to thrive (*Ehrlich et al., 2008*). Chromosome anomalies including whole-arm deletions and multiradial chromosomes have also been reported in mitogen-stimulated lymphocytes from ICF-patients. However, similar chromosome anomalies are not observed in primary tissues from affected individuals (*Ehrlich, 2003*).

Homozygosity mapping and whole-exome sequencing have separately implicated four genes in ICF syndrome: DNA Methyltransferase 3B (*DNMT3B,* ICF type-1), Zinc-finger and BTB domain containing 24 (*ZBTB24,* ICF type-2), Cell division cycle associated 7 (*CDCA7,* ICF type-3) and Helicase,

lymphoid-specific (*HELLS,* ICF type-4) (*de Greef et al., 2011*; *Thijssen et al., 2015*; *Xu et al., 1999*). Most of the described mutations in *DNMT3B* cause amino acid substitutions within the C-terminal catalytic domain, suggesting they may be hypomorphic. In contrast, the majority of mutations in *ZBTB24*, *CDCA7* and *HELLS* are predicted to cause loss of function. Mechanistically, ZBTB24, CDCA7 and HELLS are thought to converge in a singular pathway that facilitates DNMT3B access to pericentromeric DNA (*Jenness et al., 2018*; *Wu et al., 2016*).

To date, most studies of pericentromeric 5mC loss have been performed using transformed B-cell lines derived from ICF patients carrying mutations in *DNMT3B* (*Ehrlich et al., 2008*). Attempts to generate viable mouse models of pericentromeric hypomethylation through mutation of ICF genes have had limited success. Mice harboring ICF-like mutations in *Dnmt3b* exhibit some characteristics of ICF syndrome including small size and facial anomalies. However, most mice die within 24 hr of birth (*Ueda et al., 2006*). Global methylation profiles were not assessed in these mutants; but significant hypomethylation was reported at both pericentromeric repeats and retroviral sequences. Similar perinatal lethality was observed following deletion of the mouse *HELLS* orthologue. In this case, mutations were accompanied by roughly 50% reductions in 5mC, and methylation loss was noted at pericentromeres, retroviruses and some single copy sequences (*Tao et al., 2011*). Deletion of the mouse *Zbtb24* gene was reported to cause embryonic lethality; but methylation changes in these mutants have not been investigated (*Wu et al., 2016*).

Here, we describe a viable model of pericentromeric methylation loss, generated through mutation of the zebrafish *zbtb24* gene. Homozygous mutant adults exhibited key phenotypic hallmarks of ICF syndrome including hypomethylation of pericentromeric satellite repeats. Hypomethylation of these repeats was first detected in mutants at 2 weeks post fertilization (wpf) and became more severe as animals matured. This progressive methylation loss allowed us to investigate the primary consequences of pericentromeric hypomethylation in the context of a vertebrate animal. Using this model, we link derepression of transcripts from hypomethylated pericentromeres to activation of an interferon-based innate immune response, and we demonstrate that this response is mediated through the MDA5-MAVS RNA sensing machinery. Our findings provide insights into the earliest consequences of pericentromeric hypomethylation, demonstrating an unappreciated function for methylation of pericentromeric repeats in protecting against autoimmunity.

## Results

### Mutation of zebrafish *zbtb24* causes ICF syndrome-like phenotypes

The zebrafish genome encodes a single, well-conserved orthologue of ZBTB24, which we mutated using TAL effector nucleases (TALENs) (*Figure 1A* and *Figure 1—figure supplements 1* and *2*). The recovered 7.9 kb deletion allele (*zbtb24$^{mk22}$*; here after referred to as *zbtb24$^{\Delta}$*), eliminates coding sequence between exons 2 and 5 (*Figure 1B*). Animals that were homozygous for this deletion lacked detectable *zbtb24* transcripts, suggesting *zbtb24$^{\Delta}$* is a null allele (*Figure 1—figure supplement 2D*). *Zbtb24$^{\Delta/\Delta}$* embryos were born to heterozygous parents at the expected Mendelian ratios and had no obvious morphological abnormalities during the first two weeks of development (*Figure 1C*). Phenotypes that were reminiscent of ICF syndrome emerged as animals matured. Consistent with the small stature observed in ICF syndrome, by 3–4 weeks post fertilization (wpf), *zbtb24$^{\Delta/\Delta}$* mutant zebrafish were smaller than wild-type siblings raised under identical conditions, and this size reduction persisted into adulthood (*Figure 1D–F*). As adults, *zbtb24$^{\Delta/\Delta}$* mutants exhibited facial anomalies that were characterized by a quantifiable elongation of the snout (*Figure 1G–H*). We also noted evidence of hypogammaglobulinemia in the presence of normal lymphoid cell numbers, which is an immunological hallmark of ICF syndrome (*Figure 1I–J*). Significant death was noted among homozygous mutants at 4 months of age and fewer than 10% of *zbtb24$^{\Delta/\Delta}$* animals survived beyond 8 months (*Figure 1K*). Attempts to recover fertilized embryos by intercrossing or outcrossing *zbtb24$^{\Delta/\Delta}$* adults were unsuccessful, suggesting that animals were sterile (*Figure 1—figure supplement 3A*). Gonadal morphology in *zbtb24$^{\Delta/\Delta}$* mutants appeared overtly similar to wild-type siblings in histological sections (*Figure 1—figure supplement 3B–C*). However, testes size and sperm count were severely reduced in *zbtb24* mutants, providing one potential explanation for impaired male fertility (*Figure 1—figure supplement 3D–G*). Similar ICF-like phenotypes were observed in zebrafish that were homozygous for a second independently-isolated mutant allele of

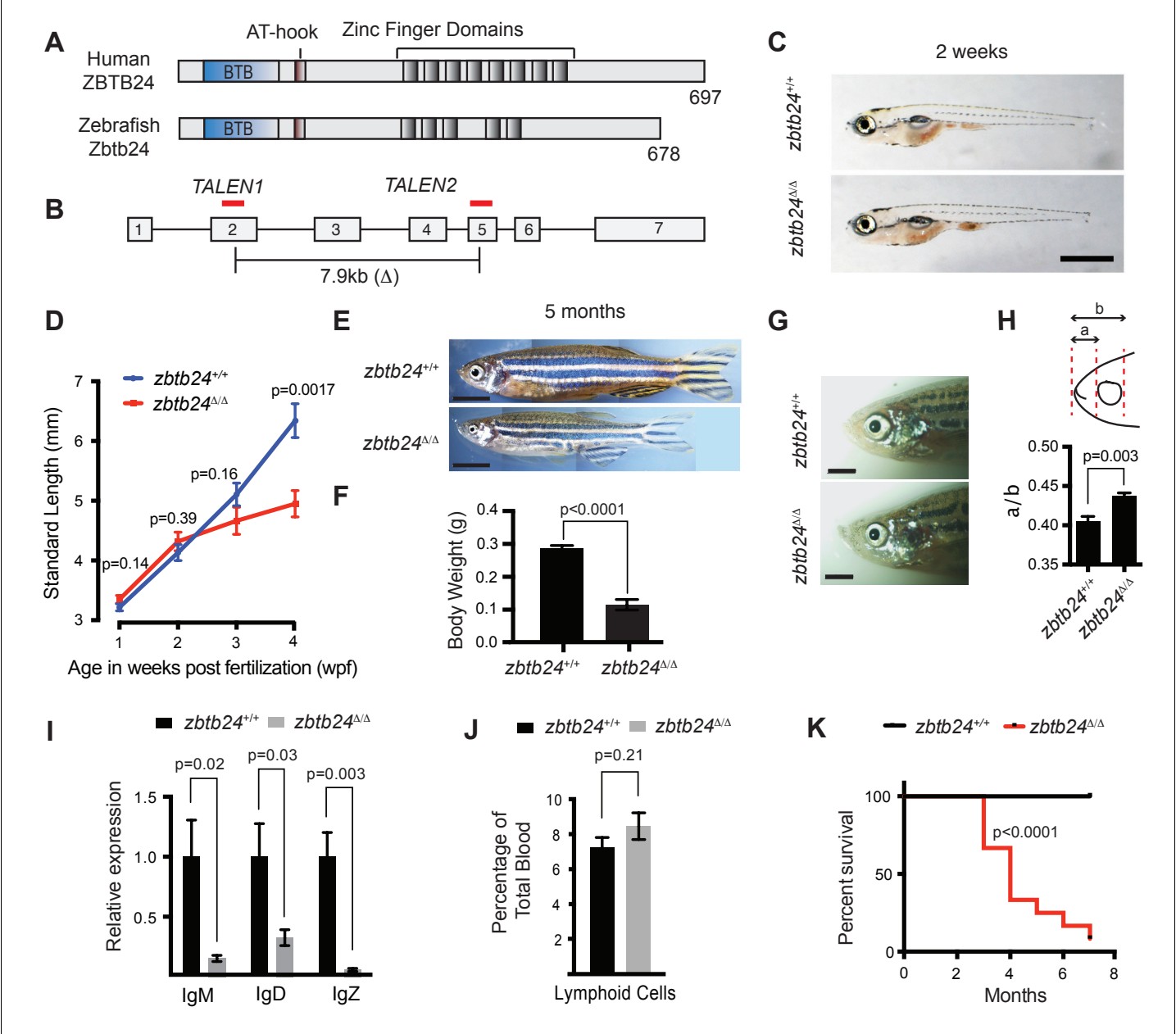

**Figure 1.** Mutation of *zbtb24* causes ICF syndrome-like phenotypes in zebrafish. (**A**) Schematic of human and zebrafish Zbtb24 proteins. The BTB/POZ domain is indicated in blue and C2H2-type zinc fingers in dark grey. (**B**) Schematic of zebrafish *zbtb24* gene. Location of TALEN target sequences are indicated in red (not to scale). Brackets indicate the region deleted by the *zbtb24^mk22(Δ)* allele. (**C**) Representative images of *zbtb24^+/+* and *zbtb24^Δ/Δ* zebrafish at 2 wpf. Scale bar: 1 mm. (**D**) Standard length measurements for *zbtb24^+/+* and *zbtb24^Δ/Δ* zebrafish at 1, 2, 3 and 4 wpf (n ≥ 6 for each group). (**E**) Representative images of *zbtb24^+/+* and *zbtb24^Δ/Δ* zebrafish at 5 months. Scale bar: 5 mm. (**F**) Average weight of *zbtb24^+/+* and *zbtb24^Δ/Δ* zebrafish at 5 months (n = 5 for each group). (**G**) Representative images of facial abnormalities in *zbtb24^+/+* and *zbtb24^Δ/Δ* adults at 6 months. Scale bar: 2 mm. (**H**) Schematic and quantification of facial abnormalities in *zbtb24^Δ/Δ* zebrafish (n = 5 for each group). (**I**) Abundance of *IgM*, *IgD* and *IgZ* transcripts in *zbtb24^+/+* and *zbtb24^Δ/Δ* zebrafish at 6 weeks post fertilization (n = 5 for each group). (**J**) Quantification of lymphoid cell populations in total blood isolated from *zbtb24^+/+* or *zbtb24^Δ/Δ* kidney marrow from adults, measured by Forward/Side scatter flow cytometry (n = 11 for each group). (**K**) Kaplan-Meier curve indicating survival among groups of *zbtb24^+/+* and *zbtb24^Δ/Δ* zebrafish (n = 12 for each group). All error bars indicate standard error of the mean (SEM).

DOI: https://doi.org/10.7554/eLife.39658.003

The following figure supplements are available for figure 1:

**Figure supplement 1.** Zbtb24 conservation in vertebrate species.

DOI: https://doi.org/10.7554/eLife.39658.004

*Figure 1 continued on next page*

*Figure 1 continued*

**Figure supplement 2.** TALEN design for introducing mutations at the endogenous *zbtb24* zebrafish gene.
DOI: https://doi.org/10.7554/eLife.39658.005
**Figure supplement 3.** Fertility and gonad analysis in *zbtb24*$^{\Delta/\Delta}$ mutants.
DOI: https://doi.org/10.7554/eLife.39658.006
**Figure supplement 4.** A second mutant allele of *zbtb24* recapitulates key features of ICF Syndrome.
DOI: https://doi.org/10.7554/eLife.39658.007

*zbtb24* (*zbtb24*$^{mk19}$) (*Figure 1—figure supplement 4*). Taken together, these findings identify *zbtb24* homozygous mutant zebrafish as a faithful animal model of ICF syndrome phenotypes.

## Progressive methylation loss at pericentromeric repeats in *zbtb24* mutants

Pericentromeric satellite type-1 (Sat1) repeats are found on all zebrafish chromosomes and comprise 5–8% of the zebrafish genome (*Phillips and Reed, 2000*). As expected, we found that Sat1 sequences from wild-type adults were resistant to digestion with the methylation sensitive restriction enzyme HpyCH4IV, indicating that these pericentromeric repeats were heavily methylated. In contrast to wildtype, Sat1 sequences from *zbtb24*$^{\Delta/\Delta}$ and *zbtb24*$^{mk19/mk19}$ mutant adults were readily digested with HpyCH4IV, indicating extensive loss of methylation at these repeats (*Figure 2A–B* and *Figure 2—figure supplement 1A*). Comparable Sat1 methylation deficiencies were observed when DNA was isolated from dissected adult brain, skin, muscle and fin, suggesting that these sequences were similarly hypomethylated in most adult somatic tissues (*Figure 2—figure supplement 1B* and *Figure 2—figure supplement 3A*). Methylation levels at Sat1 repeats appeared normal in remaining sperm extracted from *zbtb24*$^{\Delta/\Delta}$ mutant adults, suggesting methylation loss may be restricted to somatic tissues (*Figure 2—figure supplement 1C–D*).

Somewhat unexpectedly, we found that pericentromeric methylation loss in *zbtb24*$^{\Delta/\Delta}$ mutants was progressive. While extensive hypomethylation of Sat1 sequences was detected in adults lacking *zbtb24*, similar hypomethylation was not observed in mutants at 1 wpf (*Figure 2C–D*). At 2 wpf, *zbtb24*$^{\Delta/\Delta}$ mutants exhibited roughly 3-fold increases in HpyCH4IV digestion, and sensitivity to digestion became increasingly pronounced in older animals (*Figure 2C–D*). By 32 weeks, Sat1 sequences from *zbtb24* mutants exhibited a 23-fold increase in HpyCH4IV digestion compared to wildtype, suggesting a greater than 95% reduction in methylation of these repetitive sequence blocks. Histone H3 lysine nine trimethylation levels were unaffected at Sat1 sequences in *zbtb24*$^{\Delta/\Delta}$ mutant adults (*Figure 2—figure supplement 2*).

## *Zbtb24* mutants exhibit modest reductions in 5mc at non-pericentromeric sequences

To clarify whether other sequences were also hypomethylated in *zbtb24* mutants, we performed Enhanced Reduced Representation Bisulfite Sequencing (ERRBS) using genomic DNA isolated from the fins of three 6-month-old zbtb24$^{\Delta/\Delta}$ mutant adults and three wild-type siblings (*Garrett-Bakelman et al., 2015*). At this stage, Sat1 sequences from isolated fins were 20-fold more sensitive to HypCH4IV in *zbtb24* mutants compared to controls, indicating extensive loss of DNA methylation at pericentromeric repeats (*Figure 2—figure supplement 3A–B*). We then used ERRBS data to interrogate the methylation status of 979,971 non-pericentromeric CpG sites across the genome in the same tissue samples. Our analysis revealed a strong correlation between genome wide 5mC levels in wild-type and zbtb24$^{\Delta/\Delta}$ mutant adults (Pearson's correlation value of 0.928), although overall methylation levels appeared reduced by ~10% at all methylated sequence features in mutants (*Figure 2E–F* and *Figure 2—figure supplement 4*). Reductions consisted primarily of small-magnitude changes in 5mC across the genome, with only 1.3% (13,205) of examined CpG dinucleotides exhibiting methylation differences of greater than 20%. Consistent with these findings, at a threshold of 20% change (p-value<0.01), only 55 differentially methylated regions (DMRs) were identified between wild-type and zbtb24$^{\Delta/\Delta}$ adults (*Supplementary file 4*). Methylation levels at endogenous retroviruses and other transposable elements were also examined by methylation sensitive restriction digest. All tested elements were similarly resistant to digestion in *zbtb24*$^{\Delta/\Delta}$ mutant adults and

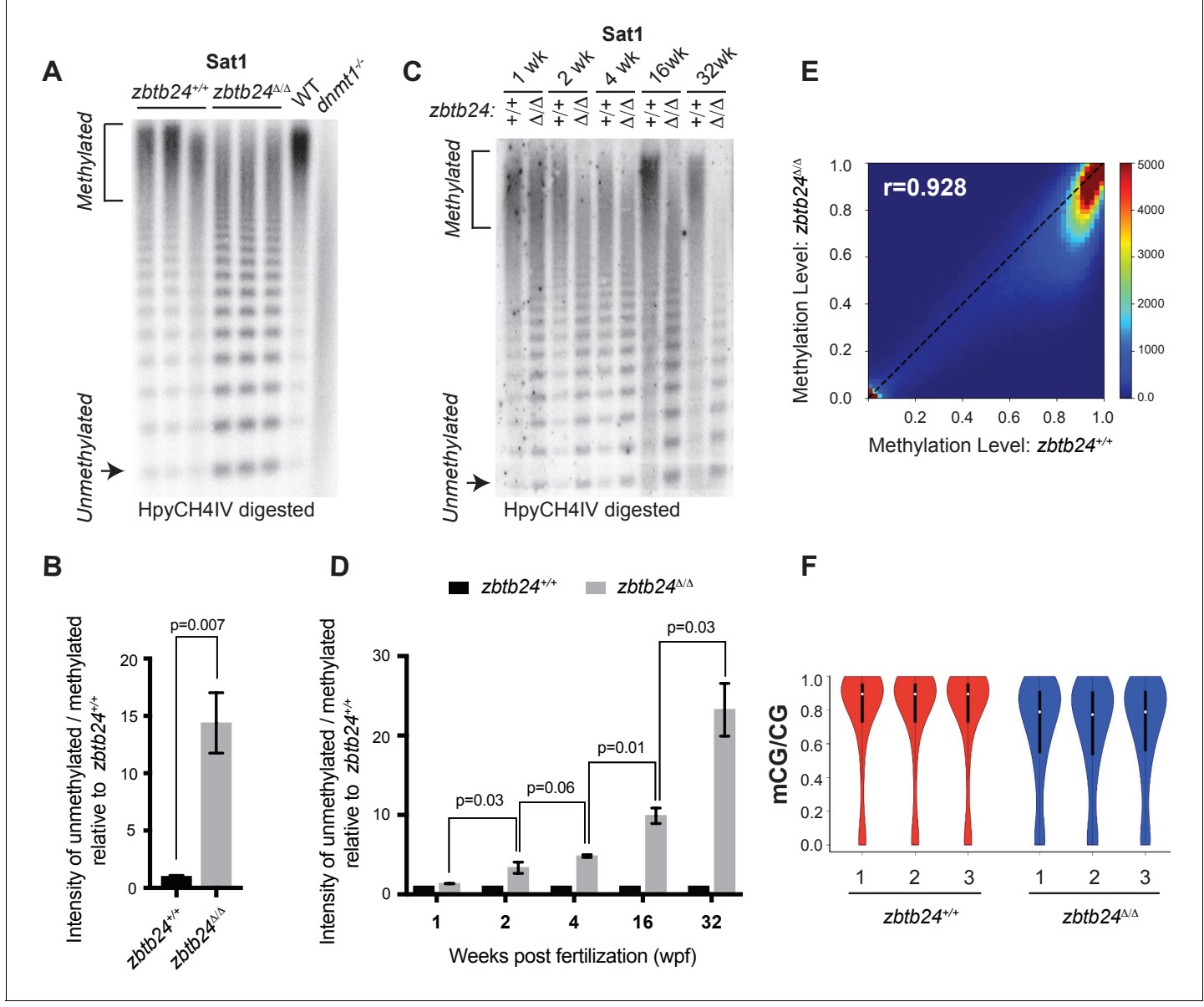

**Figure 2.** Mutation of *zbtb24* causes progressive methylation loss at pericentromeric satellite repeats. (A) Southern blot of genomic DNA digested with 5mC-sensitive restriction enzyme HpyCH4IV and probed with zebrafish Sat1 sequence. Each lane represents DNA isolated from one adult individual of the indicated genotype. DNA from *dnmt1*[−/−] zebrafish larvae at 7 days post fertilization and their phenotypically wild-type siblings (WT) provides a positive control. (B) Quantification of methylation changes at Sat1 sequences in panel A). Error bars indicate SEM from the three biological replicates. (C) Southern blot of genomic DNA digested with 5mC-sensitive restriction enzyme HpyCH4IV and probed with zebrafish Sat1 sequence. Genomic DNA was isolated from *zbtb24*[+/+] and *zbtb24*[Δ/Δ] animals at 1, 2, 4, 16 and 32 wpf as indicated. (D) Quantification of methylation changes at Sat1 sequences in panel C). Data represent averages from two independent experiments. Error bars represent the standard deviation (SD). (E) Correlation heat map of CpG methylation levels in *zbtb24*[+/+] and *zbtb24*[Δ/Δ] as assessed by ERRBS (Data reflects three biological replicates of each genotype). The density of CpGs increases from blue to dark red. (F) Violin Plots indicating overall CpG methylation levels in fins from adult *zbtb24*[+/+] and *zbtb24*[Δ/Δ] zebrafish.

DOI: https://doi.org/10.7554/eLife.39658.008

The following figure supplements are available for figure 2:

**Figure supplement 1.** *Zbtb24* mutation causes methylation loss at pericentromeric repeats.
DOI: https://doi.org/10.7554/eLife.39658.009

**Figure supplement 2.** Repressive histone modifications are unaffected in *zbtb24*[Δ/Δ] mutants.
DOI: https://doi.org/10.7554/eLife.39658.010

**Figure supplement 3.** *Zbtb24* mutants exhibit modest reductions in 5mC at non-pericentromeric sequences.
DOI: https://doi.org/10.7554/eLife.39658.011

*Figure 2 continued on next page*

*Figure 2 continued*

**Figure supplement 4.** DNA Methylation levels at different genomic classes.
DOI: https://doi.org/10.7554/eLife.39658.012
**Figure supplement 5.** Methylation at interspersed repeats is unaffected in *zbtb24* mutants.
DOI: https://doi.org/10.7554/eLife.39658.013

wild-type siblings (*Figure 2—figure supplement 5*). Collectively, these data indicate that pericentromeres are a predominant site of methylation loss in zbtb24$^{\Delta/\Delta}$ mutants.

## Mutation of *zbtb24* causes activation of innate immune response genes

To gain insights into the early consequences of methylation loss in *zbtb24* mutants, we performed transcriptome analysis on RNA isolated from wild-type and *zbtb24*$^{\Delta/\Delta}$ zebrafish at 2 wpf. At this stage, *zbtb24*$^{\Delta/\Delta}$ mutants remain morphologically indistinguishable from wildtype, but show clear hypomethylation of pericentromeric sequences. RNA-seq identified 58 genes that were downregulated by more than 2-fold in *zbtb24*$^{\Delta/\Delta}$ larvae at 2 wpf, while 119 were upregulated by 2-fold or more (*Figure 3A*). No gene enrichment signature was observed among downregulated genes. However, roughly 30% of upregulated genes were associated with activation of the innate immune system. In particular, we noted that upregulated transcripts included those associated with interferon stimulated genes (ISGs) and inflammatory cytokines (*Figure 3B*). Consistent with these observations, Gene Set Enrichment Analysis (GSEA) identified significant enrichment of genes involved in viral response, a key function of innate immune pathways (*Figure 3C*). Upregulation of ISGs was also observed in *zbtb24*$^{\Delta/\Delta}$ and *zbtb24*$^{mk19/mk19}$ mutants by qRT-PCR at 3 wpf, whereas the same genes were expressed at wild-type levels at 1 wpf (*Figure 3D–E* and *Figure 3—figure supplement 1*). No immune-related genes (and only one gene differentially upregulated in the RNA-Seq) were found within 100 kb of identified DMRs, suggesting that direct loss of methylation at these sequences was unlikely to cause the response (*Figure 2—figure supplement 3D* and *Supplementary file 4*). Consistent with previous studies, we found that global methylation depletion using the DNA methyltransferase inhibitor 5-azacytidine also resulted in upregulation of immune response genes (*Figure 3—figure supplement 2*).

## The innate immune response in *zbtb24* mutants is mediated by sensors of cytosolic RNA

The innate immune system represents an ancient defense system in which pathogen-associated molecular patterns (PAMPs) are recognized by pattern recognition receptors (PRRs). These PRRs induce signaling cascades that drive the production of interferons and other inflammatory cytokines with antiviral and immune modulating functions (*Schneider et al., 2014*). In addition to extracellular pathogens, PRRs also recognize PAMPs associated with cell-intrinsic stimuli including DNA damage, endogenous retroviral RNA and RNA-DNA hybrids (*Chiappinelli et al., 2015*; *Härtlova et al., 2015*; *Mankan et al., 2014*; *Roulois et al., 2015*).

To clarify the origin of the response in *zbtb24* mutants, we examined the major families of PRRs involved in innate immunity. These include the Toll-like receptors (TLRs), which have broad functions in detecting PAMPs, the RIG-I like receptors (RLRs), which are involved in the detection of cytosolic RNA and cGAMP synthase (cGAS), which functions as a cytosolic sensor of DNA and RNA/DNA hybrids (*Crowl et al., 2017*). Mutations in key mediator proteins required to propagate interferon signaling from each PRR family were introduced onto the *zbtb24* mutant background and we tested the effect on ISG expression. Mutations in the zebrafish orthologs of *mitochondrial antiviral-signaling protein* (*mavs*), which is an intermediate in RLR signaling and stimulator of interferon genes (sting), which is involved in *cGAS* signaling were generated using CRISPR/Cas9 technology (*Figure 4—figure supplement 1A–B*). The mutant allele of *Myeloid differentiation primary response 88* (*myd88*), which is required for signaling through most TLRs, was previously described (*van der Vaart et al., 2013*).

As in prior experiments, significant increases of the ISGs, *signal transducer and activator of transcription 1b* (*stat1b*) and *interferon regulatory factor* (*irf7*) were observed in *zbtb24*$^{\Delta/\Delta}$ larvae at 3 wpf by qRT-PCR (*Figure 4A–C*). Introduction of *myd88* or *sting* mutations had little impact on

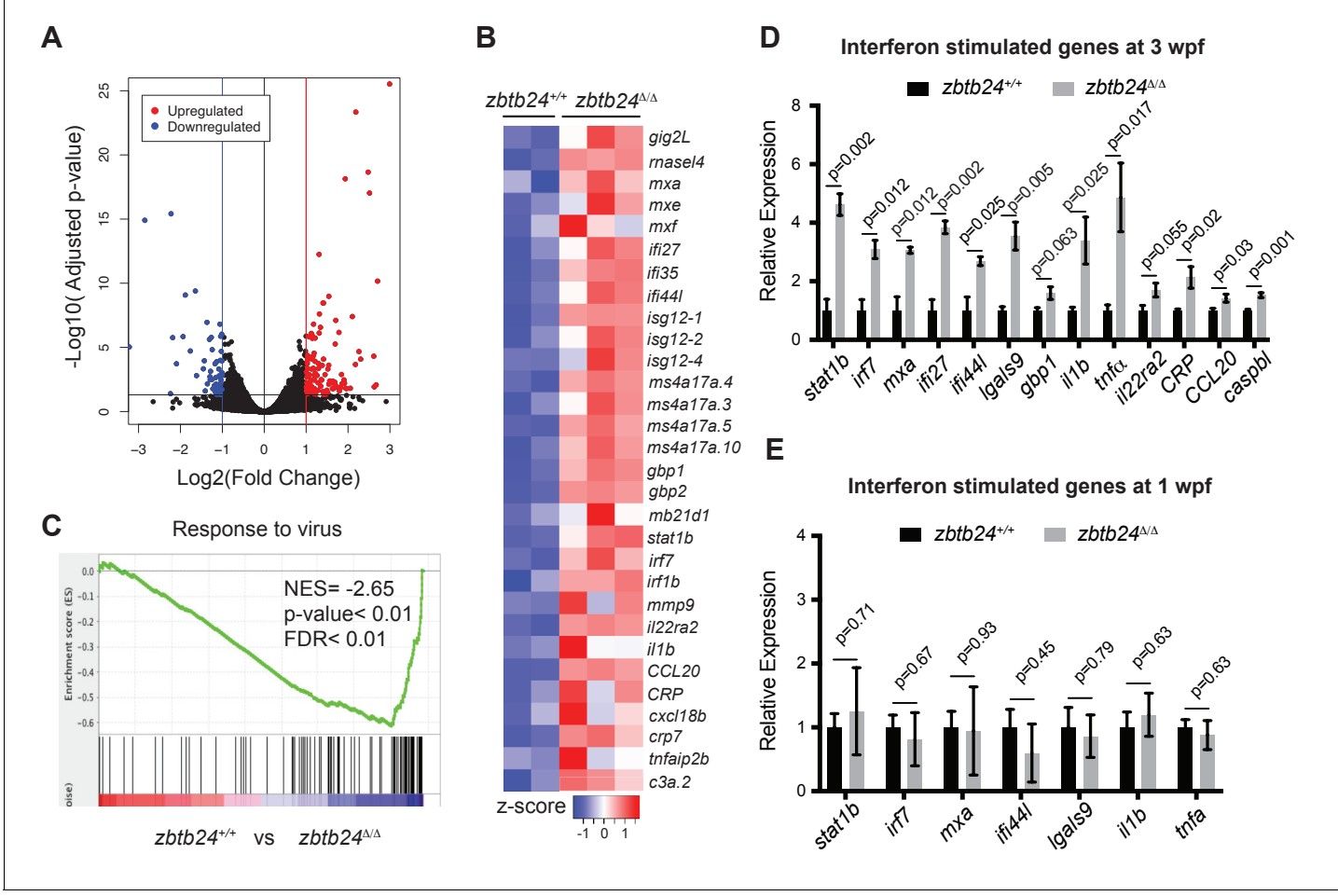

**Figure 3.** Mutation of *zbtb24* leads to activation of innate immune response genes. (A) Volcano plot representation of differential gene expression in *zbtb24*[+/+] vs *zbtb24*[Δ/Δ] zebrafish at 2 wpf. Blue and red points mark genes with >2 fold downregulation or upregulation respectively. (B) RNA-seq heatmap showing innate immune genes upregulated in *zbtb24*[Δ/Δ] mutant compared to *zbtb24*[+/+] siblings. Shown are Z-score normalized gene expression values. (C) GSEA of a set of genes involved in Response to Virus in zebrafish comparing *zbtb24*[+/+] vs *zbtb24*[Δ/Δ]. NES, normalized enrichment score; FDR, false discovery rate. (D) qRT-PCR demonstrating upregulated interferon and inflammatory response genes in *zbtb24*[Δ/Δ] mutants at 3 wpf. Expression levels are reported relative to *β-actin*. Error bars indicate SEM from at least 3 independent biological replicates with n = 8 total animals for each replicate. (E) qRT-PCR analysis reveals similar expression of interferon genes in *zbtb24*[+/+] and *zbtb24*[Δ/Δ] larvae at 1 wpf. Error bars represent SEM from at least five biological replicates.

DOI: https://doi.org/10.7554/eLife.39658.014

The following figure supplements are available for figure 3:

**Figure supplement 1.** Mutation in *zbtb24* leads to activation of innate immune response pathways.
DOI: https://doi.org/10.7554/eLife.39658.015

**Figure supplement 2.** Activation of interferon stimulated genes upon treating zebrafish embryos with 5aza-cytidine.
DOI: https://doi.org/10.7554/eLife.39658.016

expression of these ISGs, as similar transcript levels were detected in *zbtb24*[Δ/Δ] single mutant animals compared to *myd88*[hu3568/hu3568]; *zbtb24*[Δ/Δ] or *sting*[mk30/mk30]; *zbtb24*[Δ/Δ] double mutants (*Figure 4A–B*). Sustained ISG expression in these double mutants suggests limited roles for TLR and cGAS PRRs in mediating the interferon response in *zbtb24* mutants. In contrast to *myd88* and *sting*, mutation of *mavs* suppressed *stat1b* and *irf7* upregulation in *zbtb24*[Δ/Δ] mutant animals. Expression levels of *irf7* and *stat1b* were reduced 2- and 4-fold respectively in *mavs*[mk28/mk28]; *zbtb24*[Δ/Δ] double mutants when compared to *zbtb24*[Δ/Δ] single mutant zebrafish, indicating a requirement for *mavs* in

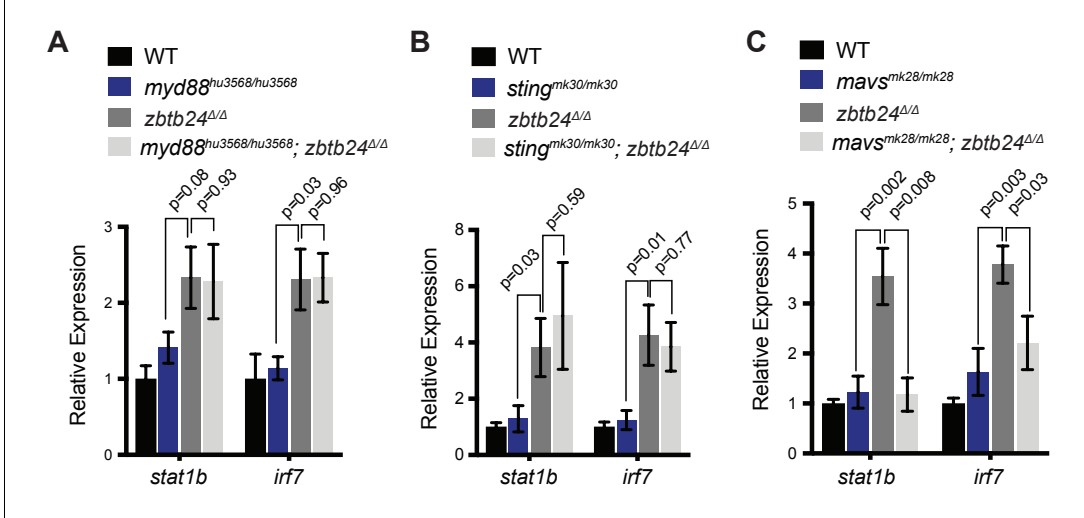

**Figure 4.** Interferon response in *zbtb24* mutants is mediated by sensors of cytosolic RNA. (**A**) Expression of interferon signaling genes *stat1b* and *irf7* in indicated genotypes at 3 wpf. n = 4 biological replicates. (**B**) Expression of the ISGs *stat1b* and *irf7* in indicated genotypes at 3 wpf. n ≥ 7 biological replicates. (**C**) Expression of interferon signaling genes *stat1b* and *irf7* in indicated genotypes at 3 wpf. n ≥ 5 biological replicates. All error bars indicate SEM.

DOI: https://doi.org/10.7554/eLife.39658.017

The following figure supplement is available for figure 4:

**Figure supplement 1.** Mutation of zebrafish orthologs of *mavs*, *sting*, and *mda5*.
DOI: https://doi.org/10.7554/eLife.39658.018

the upregulation of these ISGs (*Figure 4C*). This finding implicates RLR signaling in the activation of the innate immune system in *zbtb24* mutants and suggests a cytosolic RNA trigger for the response.

## Pericentromeric RNA transcripts are sufficient to trigger the interferon response in *zbtb24* mutants

Given known roles for DNA methylation in transcriptional repression, we next tested whether loss of methylation at pericentromeric sequence resulted in increased levels of Sat1 transcripts that could trigger the RNA mediated interferon response. Consistent with this model, strong derepression of Sat1 RNA from hypomethylated pericentromeres was noted in *zbtb24* mutant adults (*Figure 5A* and *Figure 5—figure supplement 1A*), whereas transcripts for other dispersed repetitive elements remained unchanged between mutants and wildtype (*Figure 5—figure supplement 1B*). Increases in Sat1 transcripts correlated with levels of *irf7* expression in adult zebrafish (r = 0.77), and upregulation of Sat1 transcripts coincided with the window of ISG induction during development (*Figure 5B–C*). Both sense and antisense transcripts were detected in mutants using TAG-aided sense/antisense transcript detection (TASA-TD) strand-specific PCR (*Henke et al., 2015*), suggesting the potential for derepressed Sat1 transcripts to form double stranded RNAs (*Figure 5D–E*).

To determine whether Sat1 transcripts were sufficient to activate an innate immune response, in vitro synthesized RNA corresponding to Sat1 sense and antisense transcripts were injected into wild-type embryos at the 1 cell stage. Expression of the ISGs *stat1b*, *irf7*, *irf1b* and *mxa* was then assessed at 8 hr post fertilization. Co-injection of sense and antisense Sat1 RNA was sufficient to reproducibly cause a 2 to 4-fold upregulation in expression of these ISGs, whereas combined injection of sense and antisense control transcripts encoding a fragment of zebrafish β-actin or GFP had no effect on expression of these genes (*Figure 5F* and *Figure 5—figure supplement 2*). Lower level upregulation of some, but not all ISGs was noted when sense or antisense Sat1 transcripts were individually injected into the embryo, suggesting that the response was primarily triggered by formation of Sat1 dsRNA (*Figure 5—figure supplement 2*). Collectively, these results functionally link the derepression of Sat1 transcripts to the activation of the innate immune response in *zbtb24* mutants.

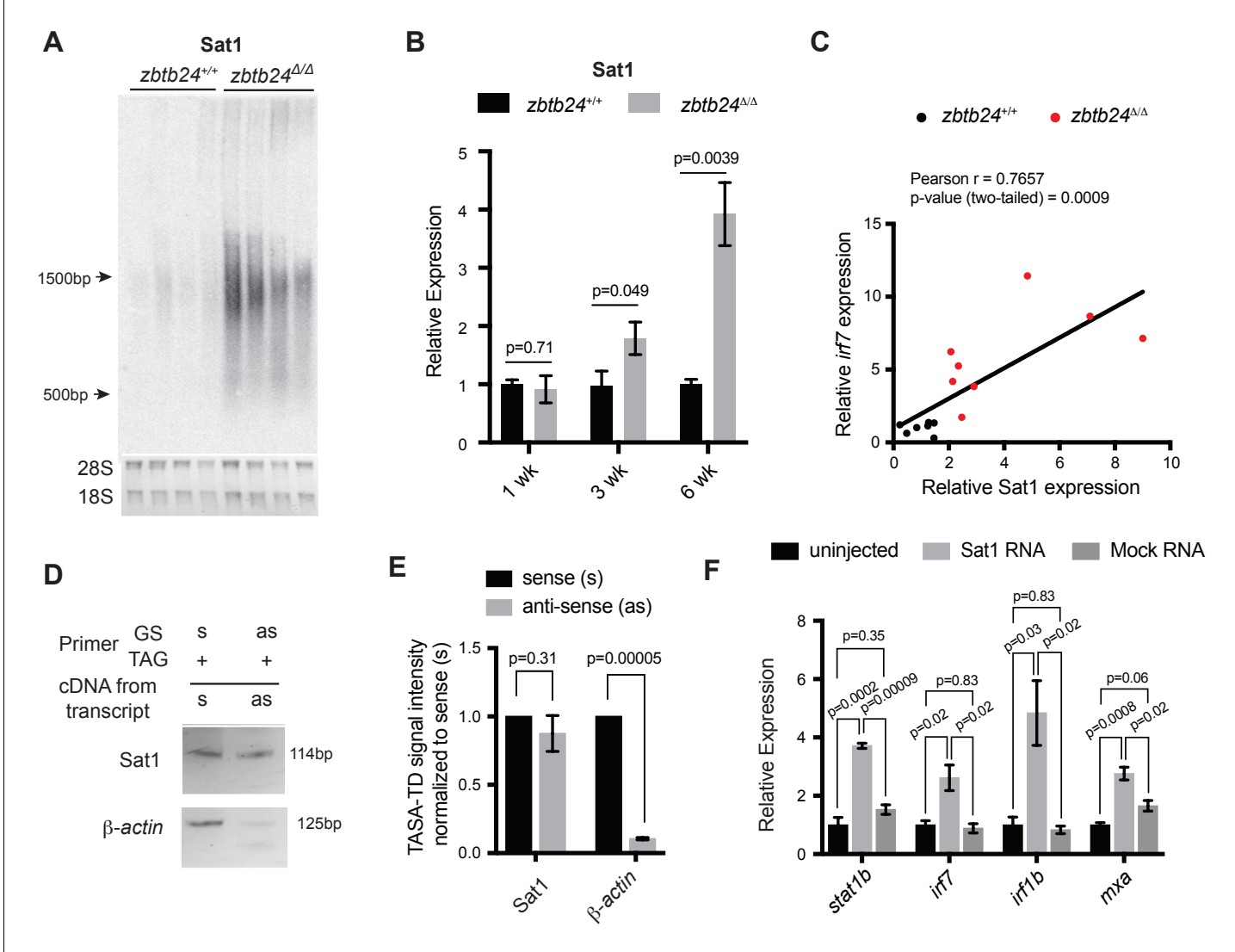

**Figure 5.** Pericentromeric transcripts are sufficient to induce the innate immune response in *zbtb24* mutants. (**A**) Northern blot analysis of Sat1 transcripts in *zbtb24*$^{+/+}$ and *zbtb24*$^{\Delta/\Delta}$ zebrafish at 6 wpf. Each lane represents a biological replicate. The lower panel represents the cropped ethidium-bromide stained gel as loading control. (**B**) qRT-PCR for Sat1 transcripts in *zbtb24*$^{+/+}$ and *zbtb24*$^{\Delta/\Delta}$ zebrafish at 1, 3 and 6 wpf. Error bars indicate SEM of at least four biological replicates in each group. (**C**) Correlation between the expression of Sat1 and *irf7* in *zbtb24*$^{+/+}$ and *zbtb24*$^{\Delta/\Delta}$ at 6 weeks (n = 15). (**D**) TASA-TD PCR amplified sense (s) and antisense (as) transcripts Sat1 (114 bp) and *β-actin* (125 bp) from first strand *zbtb24*$^{\Delta/\Delta}$ cDNA. PCR primers: gene-specific (GS); TAG. The products from TASA-TD PCR were run on the same gel, then cropped and presented. This panel is representative of two independent biological replicates. (**E**) Quantification of TASA-TD from panel **D**). Error bars indicate SD from two biological replicates. (**F**) Expression of interferon stimulated genes in wild-type embryos injected with Sat1 or control RNA encoding a similar-sized fragment of *β*-actin. 50 pg of in vitro transcribed sense and antisense transcripts were injected into wild-type zebrafish embryos at the 1 cell stage. Total RNA was extracted at 8 hr post fertilization for qRT-PCR analysis. Error bars indicate SEM from at least three biological replicates with n = 20 embryos for each biological replicate.

DOI: https://doi.org/10.7554/eLife.39658.019

The following figure supplements are available for figure 5:

**Figure supplement 1.** Mutation in *zbtb24* upregulates Sat1 transcripts but not transposons.

DOI: https://doi.org/10.7554/eLife.39658.020

**Figure supplement 2.** Effect of injecting in vitro transcribed Sat1 RNAs on expression of interferon stimulated genes.

DOI: https://doi.org/10.7554/eLife.39658.021

## The cytosolic RNA helicase MDA5 is required for the interferon response in *zbtb24* mutants

Finally, we sought to identify the specific PRR required for the interferon response in *zbtb24* mutants. The RLR family of PRRs consists of two RNA helicases that signal through Mavs: Melanoma Differentiation-Associated protein 5 (Mda5) and Retinoic acid-inducible gene I (Rig-I). Rig-I binds 5' triphosphorylated RNA molecules, whereas Mda5 has been implicated in the recognition of long double-stranded RNAs in the cytosol (*Crowl et al., 2017*). Given that 5' triphosphorylation of RNAs is a typical viral signature that is unlikely to be present on endogenous RNA transcripts, we reasoned that Mda5 was a more likely candidate for the receptor. To test the requirement for *mda5*, we generated a seven base-pair deletion in this gene that disrupted the DEAD box helicase domain (*Figure 4—figure supplement 1C*). This *mda5^mk29^* allele was then introduced onto the *zbtb24* mutant background, and expression of the ISGs *stat1b* and *irf7* was examined at 3 wpf and 6 wpf. Homozygous mutation of *mda5* was sufficient to restore *stat1b* and *irf7* expression to wild-type levels in *zbtb24*^Δ/Δ^ mutant larvae, suggesting that Mda5 is the primary PPR required for the response (*Figure 6A and B*). This requirement was further validated by RNA-seq, which revealed that a broad panel of ISGs that showed elevated expression in *zbtb24* single mutants were no longer upregulated in *mda5*^mk29/mk29^; *zbtb24*^Δ/Δ^ double mutants (*Figure 6C*).

Taken together, these results support a model in which derepression of transcripts from hypomethylated pericentromeres triggers activation of the innate immune system through the Mda5/Mavs viral RNA recognition pathway (*Figure 6D*). These findings identify roles for pericentromeric RNA as a trigger of autoimmunity and reveal important functions for pericentromeric methylation in suppressing the generation of these immunostimulatory transcripts. Based on these results, we propose that induction of the innate immune system is one of the earliest in vivo consequences of pericentromeric methylation loss.

## Discussion

In this study, we describe a viable animal model of ICF syndrome which recapitulates key phenotypic hallmarks of the disease including slow growth, facial anomalies, immunoglobulin deficiencies and reduced lifespan. Given that previous attempts to model ICF syndrome have resulted in perinatal or embryonic lethality (*Geiman et al., 2001*; *Ueda et al., 2006*; *Wu et al., 2016*), this zebrafish model provides an important new resource for understanding ICF disease etiology during juvenile and adult life stages. In particular, *zbtb24* mutant zebrafish will be useful for understanding phenotypes such as immunoglobulin deficiency, which have not been observed in mouse models and are difficult to study in cell culture systems.

As in ICF syndrome, *zbtb24* mutant adult zebrafish exhibited extensive loss of methylation at pericentromeric sequences. For highly repetitive sequences, methylation sensitive restriction digest followed by Southern blot remains the most effective way to assess methylation levels. By this approach, we observed increases in HpyCH4IV digestion that are consistent with up to 95% reductions in methylation at Sat1 pericentromeric repeats in *zbtb24* mutants. While similar hypomethylation was observed in all adult somatic tissues that we examined, we unexpectedly observed that methylation levels in sperm from *zbtb24* mutants and wildtype animals appeared comparable. This finding raises the possibility that different pathways act to control pericentromeric methylation in germ and somatic cells.

Methylation levels at pericentromeric Sat1 sequences could not be quantified by ERRBS, as this technique relies on MspI restriction digest to enrich for CpG containing sequences, and zebrafish Sat1 repeats are lacking in this restriction site. Nonetheless, ERRBS analysis suggested that the general methylation landscape in human ICF syndrome and in *zbtb24* mutant zebrafish is similar. Methylome analysis of primary blood from ICF patients identified methylation changes of greater than 20% at roughly 3% of examined CpG dinucleotides. Significant changes in methylation of retroviruses and other dispersed repeats were not observed in these patients (*Velasco et al., 2018*). Consistent with these findings, our ERRBS analysis revealed methylation changes of greater than 20% at roughly 1.3% of assayed CpG dinucleotides and found methylation of dispersed repeats to be similar between wildtype and in *zbtb24* mutant zebrafish. The low-level methylation changes outside of the pericentromeres observed in ICF syndrome and our mutants raise the possibility that *zbtb24* may have additional modest roles in maintaining methylation at non pericentromeric sequences. One

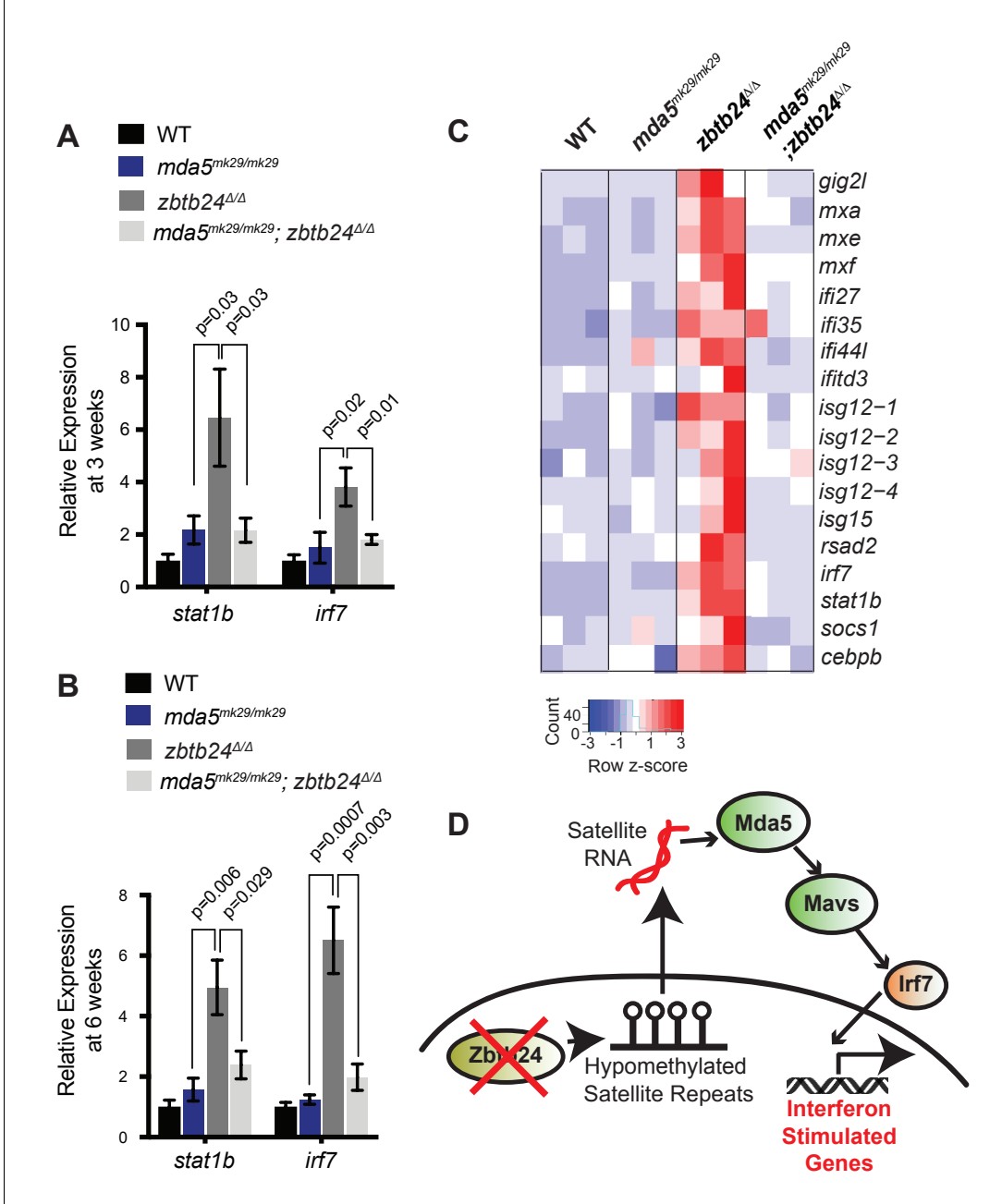

**Figure 6.** Mutation of cytosolic RNA receptor Mda5 mitigates the interferon response in $zbtb24^{\Delta/\Delta}$ zebrafish. (A) Expression of interferon signaling genes *stat1b* and *irf7* in indicated genotypes at 3 wpf. n ≥ 7 biological replicates. (B) Expression of interferon signaling genes *stat1b* and *irf7* in indicated genotypes at 6 wpf. n = 6 biological replicates. (C) RNA-seq heatmap of interferon stimulated genes upregulated in $zbtb24^{\Delta/\Delta}$ zebrafish and rescued in $mda5^{mk29/mk29}$; $zbtb24^{\Delta/\Delta}$ zebrafish at 3 wpf. Shown are Z-score normalized gene expression values. (D) Model for the activation of interferon response in *zbtb24* mutants. Loss of Zbtb24 function causes hypomethylation of pericentromeric Sat1 repeats, which leads to derepression of associated Sat1 transcripts. These pericentromeric transcripts are recognized by the RNA helicase Mda5 which signals through Mavs and Irf7 to upregulate ISGs. Autoregulatory feedback implicates *irf7* as both an ISG and a key downstream effector of Mda5/Mavs signaling.

DOI: https://doi.org/10.7554/eLife.39658.022

The following figure supplement is available for figure 6:

**Figure supplement 1.** Expression of ICF genes *cdca7* and *hells* in zbtb24 mutants (A) Expression levels of *cdca7* and *hells* in the RNA Seq data set reported in **Figure 6C** (n = 3 biological replicates for each group).

DOI: https://doi.org/10.7554/eLife.39658.023

important caveat of ERRBS analysis is that CpG poor sequences can be under represented, leaving open the possibility that additional DMRs in CpG poor regions of the genome were overlooked by our approach.

The progressive loss of 5mC we observe in somatic tissues between larval and adult stages implicates Zbtb24 in regulating the long-term maintenance of methylation at pericentromeric repeats. We are unaware of any developmental or methylation milestones that can account for the onset of hypomethylation around 2 wpf. Rather, we speculate that the onset of methylation loss at this stage partly reflects the need to deplete maternally loaded *zbtb24* prior to unmasking of the *zbtb24* mutant phenotype and partly reflects the culmination of minor methylation losses due to lower fidelity maintenance over many rounds of cell division. We note that the onset of ICF-like growth defects in *zbtb24* mutant zebrafish emerged in the weeks following Sat1 methylation loss. In at least one case of ICF syndrome type 2, growth reductions and immunodeficiency were also reported to develop with age, raising the possibility that similar progressive methylation loss may impact ICF etiology in humans (*von Bernuth et al., 2014*). It is also possible that Zbtb24 functions in both maintenance and establishment of pericentromeric methylation, but that requirements for establishment are masked by maternally deposited RNA in *zbtb24* mutant zebrafish lines. Unfortunately, *zbtb24* homozygous mutant zebrafish are sterile, preventing the generation of the maternal-zygotic mutants required to address this question.

Previous studies have suggested that ZBTB24 is a transcription factor that may act to regulate DNA methylation through transcriptional control of the ICF gene CDCA7 (*Wu et al., 2016*). Consistent with this model, we observe near complete loss of cdca7 expression in *zbtb24* mutants in our RNA-seq data set and by qRT-PCR (*Figure 6—figure supplement 1*). A more recent study in cultured human cells proposed that ZBTB24 binding might be directly involved in recruiting DNMT3B to promote gene body methylation through recognition of AGGTCCTGGCAG motifs in human cells (*Thompson et al., 2018*). Analysis using Find Individual Motif Occurrences (FIMO) (*Grant et al., 2011*), did not reveal this motif in the promoter or gene body of zebrafish *cdca7* or at Sat1 sequences.

In the current study, we take advantage of the progressive Sat1 methylation loss in *zbtb24* mutants to identify activation of interferon signaling as one of the earliest in vivo consequences of pericentromeric hypomethylation. This phenotype cannot be attributed to defects in adaptive immunity, as the zebrafish adaptive immune system is not functional until roughly 4 wpf (*Trede et al., 2004*). Induction of an interferon response has been reported in the context of global hypomethylation in cancer cell lines treated with the DNA methyltransferase inhibitor 5-azacytidine and in zebrafish mutated for the maintenance DNA methyltransferase machinery (*Chernyavskaya et al., 2017*; *Chiappinelli et al., 2015*; *Roulois et al., 2015*). In each of these cases induction of the interferon response was attributed to massive derepression of endogenous retroviral elements.

Our results are distinguished from these earlier studies in that we identify hypomethylation of pericentromeric sequences and subsequent derepression of associated satellite transcripts as a previously unappreciated trigger of innate immunity. Immunostimulatory motifs have been noted in pericentromeric RNAs derived from mouse and humans, and transcripts derived from these repeats have been observed in *p53* null mouse fibroblasts following global methylation loss (*Leonova et al., 2013*; *Tanne et al., 2015*). However, while these studies suggest the potential for pericentromeric hypomethylation to drive an interferon response in diverse vertebrate species, experimental evidence in support of this model has been lacking. Here we demonstrate a causative link between derepression of pericentromeric RNAs and the interferon response, and identify a requirement for Mda5/Mavs in mediating the response. Our findings suggest that aberrant Sat1 transcripts derived from pericentromeric repeats trigger this response, and that these transcripts may mimic features of double stranded RNA viruses in the cytosol. This finding raises the possibility that this pathway may also recognize additional endogenous RNAs that lack viral origin.

While mutation of *mda5/mavs* rescued the interferon response in *zbtb24* mutants, *mda5/mavs* mutation had little impact on other ICF phenotypes observed in *zbtb24* mutants. Therefore, we find it unlikely that the interferon response drives ICF etiology. Rather this response represents an additional consequence of pericentromeric hypomethylation. Hypomethylation of pericentromeric sequences is compatible with human viability and is observed in abnormal cell contexts including cancer and senescence. Massive increases in pericentromeric transcripts and upregulation of interferon genes have both been noted in cancer (*Cheon et al., 2014*; *Ting et al., 2011*). Our data raise

the possibility that pericentromeric hypomethylation and subsequent derepression of associated RNAs represents an important but underappreciated trigger of autoimmunity in a variety of disease states.

## Materials and methods

### Zebrafish husbandry

Zebrafish husbandry and care were conducted in full accordance with animal care and use guidelines with approval by the Institutional Animal Care and Use Committees at Memorial Sloan Kettering Cancer Center and the University of Georgia. Zebrafish were raised under standard conditions at 28° C. Wild-type lines were of the AB background. All mutant alleles are summarized in *Supplementary file 1*.

### TALEN and CRISPR mutagenesis

TALEN sequences were selected using Targeter 2.0 software (*Doyle et al., 2012*). TAL repeat assembly was achieved using the Golden Gate assembly method, and assembled repeats were integrated into the GoldyTALEN scaffold (*Bedell et al., 2012*; *Cermak et al., 2011*). Assembled vectors served as templates for in vitro mRNA transcription using the T3 mMessage mMachine kit (Ambion) according to manufacturer's instructions. 50–100 pg mRNA was injected into wild-type embryos at the one-cell stage. Injected embryos were raised to adulthood and F1 progeny were screened for germline transmission of mutations as previously described (*Li et al., 2015*). Primers used for detection of mutations and subsequent genotyping are included in *Supplementary file 1*.

Target selection for CRISPR/Cas9 mediated mutagenesis was performed using CHOPCHOP (*Labun et al., 2016*). sgRNA templates were generated either by cloning into pT7-gRNA as described by *Jao et al. (2013)* or using the oligo-based approach described in *Gagnon et al., 2014* and *Burger et al. (2016)*. All template oligos are listed in *Supplementary file 3*. sgRNAs were in vitro transcribed from their respective templates using T7 RNA polymerase (Promega) as per manufacturer protocol. Cas9 RNA was in vitro transcribed from the pT3TS-nls-zCas9-nls plasmid (*Jao et al., 2013*) using the T3 mMessage mMachine Kit (Ambion). For mutagenesis, 200–400 ng of sgRNA and ~500 ng of Cas9 mRNA were co-injected into wild-type embryos at the one-cell stage. Injected embryos were raised to adulthood, and F1 progeny were screened for germline transmission of mutations as previously described (*Li et al., 2015*). Primers used for detection of mutations and subsequent genotyping are included in *Supplementary file 1*.

### Zebrafish imaging and length measurements

All bright field imaging of zebrafish larvae and adult was performed using Olympus MVX10 with CellSens Standard software. Standard-length was documented using ImageJ as defined in *Parichy et al. (2009)*. Photoshop (Adobe) adjustments to brightness and contrast were equally applied to all images of whole zebrafish in order to improve visualization.

### FACS analysis of whole kidney marrow

Adult zebrafish at 6 months were sacrificed with a combination of tricaine (Sigma-Aldrich, CAS number 886-86-2) and rapid chilling. Whole kidneys were dissected using forceps and placed in 0.9 × PBS/5% FCS. Manual disaggregation using a P1000 pipette resulted in single cell suspensions. Cells were filtered over a 40 µm nylon mesh filter, and resuspended in PBS/FCS to give a final concentration of 100,000 cells/µl. FACS sorting of single cells were analyzed for forward/side scatter profiles. FACS data were analyzed using FloJo software.

### Histology

For Hematoxylin and Eosin (H and E) staining, adult zebrafish were fixed in 10% Neutral Buffered Formalin for 48 hr. Zebrafish were then decalcified in 0.5 M EDTA for 24 hr. After decalcification, fish were incubated overnight in 70% Ethanol before embedding in paraffin blocks. Sections were stained with H and E according to standard procedures.

## Sperm count

Adult zebrafish at 8 months were sacrificed with a combination of tricaine and rapid chilling. Whole testis was dissected using forceps and crushed in 100 ul of PBS. For determining sperm-count, sperm samples were diluted 1:20 for each fish. 10 ul of the diluted sample was then loaded onto a hemocytometer (Bright-Line, Hauser Scientific) for counting. The volume over the central counting area is 0.1 mm$^3$ or 0.1 microliter. Average number of sperm counted over the central counting area was multiplied by 10000 to obtain the number of sperm/ml of the diluted sample. The obtained value was multiplied by the dilution factor to obtain the final sperm count.

## DNA methylation analysis

For Southern blot analysis, 1 μg of purified total genomic DNA was digested with the indicated methylation sensitive restriction enzyme, fractionated by electrophoresis through a 0.9% agarose gel and transferred to nylon membrane. Sperm DNA was isolated from sperm samples collected by crushing dissected testes in PBS. Probes were PCR amplified using primers in *Supplementary file 2* and radiolabeled with $^{32}$P-dCTP using Rediprime$^{TM}$ II Random Prime Labelling System (Amersham) according to manufacturer protocol. Hybridization signals were imaged and analyzed using a Typhoon phosphorimager (GE Life Sciences). Signal intensities were measured using ImageJ. Methylation changes at Sat1 was quantified as a ratio of the intensity of the unmethylated/methylated blot regions as indicated in the respective blot.

HypCH4IV was selected for Sat1 methylation analysis over the more traditional MspI/HpaII iso-schizomer pair because Sat1 sequences lack the CCGG sites that are recognized by these enzymes.

## Chromatin immunoprecipitation (ChIP)

ChIP was performed as described in Lindeman et al. 2009 with modifications. Briefly, zebrafish juveniles at 1 month were euthanized using tricaine. Chromatin was prepared from euthanized fish by lysing flash frozen samples using an automated pulverizer (Covaris) and crosslinking using 1% Formaldehyde for 5 mins. Chromatin shearing was performed using a Covaris S220 sonicator using the following conditions: 1 ml tubes with total chromatin from each fish in buffer containing 1% SDS were sonicated using peak intensity power of 140, duty factor of 5.0 and 200 cycles per burst, for 14 min for *zbtb24*$^{+/+}$ and 6 min for *zbtb24*$^{Δ/Δ}$. Shearing was monitored using 1% agarose gel. To provide standardized input for each ChIP experiment, chromatin was diluted to A260 = 0.2. For each ChIP, 2 μg antibody per 10 μl Dynabeads and 100 μl chromatin was incubated overnight at 4°C. Following antibodies were used in this study: anti-H3K9me3 antibody (abcam ab8898), anti-H3K27me3 (Millipore 07–449), anti-H3 (abcam 1791) and IgG control (abcam ab15008). After elution, ChIP DNA and input controls were purified using QIAquick PCR purification kit (Qiagen). Eluted DNA was analyzed by qPCR using primers targeting Sat1 (*Supplementary file 2*).

## Enhanced Reduced Representation Bisulfite Sequencing (ERRBS)

50 ng of high quality genomic DNA was prepared from fin tissue from 6-month-old adult zebrafish DNA was digested with MspI and bisulfite converted using the EZ DNA methylation kit (zymo) as in *Garrett-Bakelman et al. (2015)*. Bisulphite conversion rates (calculated using non-CpG methylation conversion rates) ranged from 99.6% to 99.7% for all samples (*Figure 2—figure supplement 2C*). Amplified libraries were sequenced on the Hiseq2500 platform using a minimum of single-read 51 cycles. ERRBS data were filtered for sequence adapters, limited to the first 29 bp of the read (*Boyle et al., 2012*), and mapped to the zebrafish genome (danRer7) using BSmap (v 2.90) (*Xi and Li, 2009*). Other than limiting to the first 29 bp all other BSmap parameters were the defaults. Methylation scores were calculated as the number of unconverted reads divided by the number of total reads at each CpG site. DMRs were called as described in *Park and Wu (2016)*. DMRs with at least a 0.2 change in methylation were determined using DSS (delta = 0.2, p.threshold = 0.01). CallDMR function in DSS was used with default parameters except for p.threshold and delta as specified. Sat1 sequences are deficient in MspI sites, and are therefore not included in ERRBS data.

## RNA expression analysis

For qRT-PCR, total RNA was isolated using Trizol (Invitrogen) and precipitated with isopropanol. RNA used for assaying expression of repeat sequences subsequently was treated with DNase using

TURBO DNA-*free* Kit (Ambion) prior to analyses. RNA was converted to cDNA using GoScript Reverse Transcriptase Kit (Promega) and Real Time PCR was performed using an Applied Biosystems 7500 PCR Machine. Analysis was performed using the $2^{-\Delta\Delta Ct}$ method, with relative mRNA levels of all transcripts normalized to β-actin1 or 18S. All primer sequences are listed in *Supplementary file 2*.

For Northern blot analysis, total RNA was extracted with using Trizol (Invitrogen). 2 µg of RNA was subjected to electrophoresis on 1% agarose gel and transferred to Amersham Hybond-N+ memmbrane (GE Healthcare). The membrane was probed with $^{32}$P-dCTP radiolabeled Sat1 DNA probe at 42°C. Hybridization signals were imaged and analyzed using a Typhoon phosphorimager (GE Life Sciences).

TAG-aided sense/antisense transcript detection (TASA-TD) strand-specific PCR was performed as described by (*Henke et al., 2015*). Oligos used are listed in *Supplemental file 3*.

## Transcriptome sequencing

After RiboGreen quantification and quality control by Agilent BioAnalyzer, 500 ng of total RNA underwent polyA selection and TruSeq library preparation according to instructions provided by Illumina (TruSeq Stranded mRNA LT Kit), with 8 cycles of PCR. Samples were barcoded and run on a HiSeq 2500 High Output in a 50 bp/50 bp paired end run, using the TruSeq SBS v4 Kit (Illumina). An average of 45.3 million paired reads was generated per sample. The percent of mRNA bases averaged 62.8%.

For single-mutant RNA-seq analysis presented in *Figure 3*, reads were mapped to the Zebrafish genome (danRer7) using the rnaStar aligner v2.5.0a (*Dobin et al., 2013*). We used the two-pass mapping method outlined in *Engström et al. (2013)*. The first mapping pass used a list of known annotated junctions from Ensemble. Novel junctions found in the first pass were then added to the known junctions and a second mapping pass was done (on the second pass the RemoveNoncanoncial flag was used). Expression counts (counts per million, cpm) were computed from the mapped reads using HTSeq v0.5.3 (*Anders et al., 2015*) and Ensemble D.rerio v79 gene annotations. Normalization and differential expression was performed using DESeq (*Anders and Huber, 2010*).

For RNA-seq analysis presented in *Figure 6*, raw RNA-seq FASTQ reads were trimmed for adapters and preprocessed to remove low-quality reads using Trimmomatic v0.33 (arguments: LEADING:3 TRAILING:3 MINLEN:36) (*Bolger et al., 2014*) prior to mapping to the *Danio rerio* GRCz10 reference genome assembly. Reads were mapped using TopHat v2.1.1 (*Kim et al., 2013*) supplied with a reference General Features File (GFF) to the *Danio rerio*GRCz10 reference genome assembly, and with the following arguments: -i 10 -I 5000 –library-type fr-firststrand. Gene expression was estimated using Cuffquant (a tool from Cufflinks v2.2.1), with following arguments –library-type fr-firststrand. Expression level were normalized in FPKM units by Cuffnorm (a tool from Cufflinks v2.2.1), with following arguments –library-type fr-firststrand.

## 5-aza-dC treatment

Zebrafish embryos were treated with 5-aza-dC (Sigma-Aldrich) to the final concentration of 25 uM or 50 uM within the first 2 hr post fertilization, when zebrafish are sensitive to 5-aza-dC treatments as described in *Martin et al. (1999)*. At 24hpf, total RNA was collected for expression analysis. At 24hpf, genomic DNA was also collected and digested with methylation sensitive enzyme, HpaII, to test for global DNA hypomethylation.

## RNA synthesis and injections

Sat1 RNA and control RNAs were in vitro transcribed using Riboprobe in vitro transcription systems (Promega). Oligos to amplify the DNA template for in vitro transcription are included in *Supplementary file 3*. Sense and anti-sense transcripts were transcribed in vitro using the T3 and T7 RNA polymerases respectively. RNA was purified illustra MicroSpin G-50 Columns (GE Healthcare) and 50 ng of sense and antisense RNA was co-injected into zebrafish embryos at the 1 cell stage.

## Statistical analysis

The Student unpaired 2-tailed t-test was used for statistical analysis unless specified otherwise. Statistical analysis was performed using GraphPad PRISM software.

### Accession number

All ERRBS and RNA-Seq data reported in this paper have been deposited in GEO under the accession GSE116360.

## Acknowledgements

This research was supported by a grant from the National Institutes of Health (R01GM110092) to MGG. We thank the Goll laboratory for helpful discussions and critical reading of the manuscript and Kellee Siegfried-Harris (UMass, Boston) for advice on gonadal sections. We acknowledge the use of the Integrated Genomics Operation Core of MSKCC, funded by the NCI Cancer Center Support Grant (CCSG, P30 CA08748), Cycle for Survival, and the Marie-Josée and Henry R Kravis Center for Molecular Oncology. We also acknowledge the use of the Bioinformatics core of MSKCC for support with sequence analysis. ERRBS was performed in the Weill Cornell Medicine Epigenomics Core Facility.

## Additional information

### Funding

| Funder | Grant reference number | Author |
|---|---|---|
| National Institutes of Health | R01GM110092 | Mary Goll |

The funders had no role in study design, data collection and interpretation, or the decision to submit the work for publication.

### Author contributions

Srivarsha Rajshekar, Conceptualization, Resources, Data curation, Formal analysis, Validation, Investigation, Visualization, Methodology, Writing—original draft, Writing—review and editing; Jun Yao, Paige K Arnold, Sara G Payne, Yinwen Zhang, Teresa V Bowman, Formal analysis, Investigation; Robert J Schmitz, Formal analysis, Investigation, Writing—review and editing; John R Edwards, Formal analysis, Visualization, Writing—original draft, Writing—review and editing; Mary Goll, Conceptualization, Resources, Supervision, Funding acquisition, Validation, Methodology, Project administration, Writing—review and editing

### Author ORCIDs

Srivarsha Rajshekar (iD) https://orcid.org/0000-0002-5224-5531
Sara G Payne (iD) http://orcid.org/0000-0002-3572-9112
Mary Goll (iD) http://orcid.org/0000-0001-5003-6958

### Ethics

Animal experimentation: This study was performed in strict accordance with the recommendations in the Guide for the Care and Use of Laboratory Animals of the National Institutes of Health. All of the animals were handled according to approved institutional animal care and use committee (IACUC) protocols of Memorial Sloan Kettering Cancer Center (MSKCC- 10-08-014) and the University of Georgia (UGA A2017 05-021). The protocol was approved by the Committee on the Ethics of Animal Experiments of MSKCC and UGA.

### Decision letter and Author response

Decision letter https://doi.org/10.7554/eLife.39658.032
Author response https://doi.org/10.7554/eLife.39658.033

## Additional files

### Supplementary files
• Supplementary file 1. List of mutant alleles.
DOI: https://doi.org/10.7554/eLife.39658.024
• Supplementary file 2. List of primers.
DOI: https://doi.org/10.7554/eLife.39658.025
• Supplementary file 3. List of Oligos (5'———–3').
DOI: https://doi.org/10.7554/eLife.39658.026
• Supplementary file 4. List of DMRS.
DOI: https://doi.org/10.7554/eLife.39658.027
• Transparent reporting form
DOI: https://doi.org/10.7554/eLife.39658.028

### Data availability
Sequencing data have been deposited in GEO under accession code GSE116360. All data generated or analyzed during this study are included in the manuscript and supporting files.

The following dataset was generated:

| Author(s) | Year | Dataset title | Dataset URL | Database and Identifier |
|---|---|---|---|---|
| Rajshekar S, Edwards JR, Goll MG | 2018 | Pericentromeric hypomethylation elicits an interferon response in an animal model of ICF syndrome | https://www.ncbi.nlm.nih.gov/geo/query/acc.cgi?acc=GSE116360 | NCBI Gene Expression Omnibus, GSE116360 |

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
