## [Decision Letter]

[**Editorial note:** This article has been through an editorial process in which the authors decide how to respond to the issues raised during peer review. The Reviewing Editor's assessment is that all the issues have been addressed.]

Thank you for submitting your article "Pericentromeric hypomethylation elicits an interferon response in an animal model of ICF syndrome" for consideration by *eLife*. Your article has been reviewed by three peer reviewers, including Deborah Bourchis as the Reviewing Editor and Reviewer #1, and the evaluation has been overseen by Jessica Tyler as the Senior Editor. The following individuals involved in review of your submission have also agreed to reveal their identity: Ozren Bogdanovic (Reviewer #2); Albert Jordan (Reviewer #3).

The Reviewing Editor has highlighted the concerns that require revision and/or responses, and we have included the separate reviews below for your consideration. If you have any questions, please do not hesitate to contact us.

Below, you will find a list of the comments from the reviewers that have not been modified because we believe that you will find it useful to consider all the points. As you will see, the three reviewers are overall positive and recognize the originality and experimental quality of the work.

Please address the comments and revise your text accordingly. Limited additional experimentation is required. Defining the separate effect of sense and antisense sat1 RNAs on the interferon response was raised by two reviewers and should be included along with ds RNA injection. We also recommend to provide more insights into the fertility and germline DNA methylation phenotypes of the *Zbtb24* KO animals.

Separate reviews (please respond to each point):

*Reviewer #1:*

The ICF syndrome is an autosomal recessive disease associated with a constitutive DNA methylation defect at pericentromeric satellite DNA. Patients present with severe immunodeficiency and facial anomalies. Although being very rare (around 100 patients worldwide), the underlying genetic causes are variable: at least four genes have been implicated – a DNA methyltransferase (DNMT3B), a zinc-finger containing factor (ZBTB24), an helicase (HELLS) and a cell cycle regulator (CDCA7). Studying this disease has a double interest: dissecting the DNA methylation pathway, and addressing the impact of defective pericentrometic heterochromatin regulation on development and immunity functions.

The authors report here the first zebrafish model of ICF, through the constitutive loss of function of *Zbtb24*. First, this model recapitulates the typical pericentromeric hypomethylation, immunological defects and facial anomalies of ICF syndrome. Second, the authors report a new molecular component of the *Zbtb24*-related immunological phenotype: hypomethylation of the pericentrometic heterochromatin results in the production of immunostimulatory sat1 repeat transcripts, which trigger an interferon response through a viral RNA recognition pathway. This provides the novel concept that the earliest consequence of pericentromeric loss is an induction of the innate immune system. These are very interesting findings, even if we cannot conclude as to whether tis response is involved in the ICF pathology.

The manuscript is clear and concise, and the conclusions well supported by the data. The genetic approaches in particular are very solid.

I have a few requests for the authors though, to improve the clarity and breadth of the findings.

1) It has been shown in mammals that *Zbtb24* controls the expression of *Cdca7*, whose mutations are also involved in the ICF syndrome. Is it also the case in zebrafish? Could the authors look in their RNA seq data from *Zbtb24* KO the expression pattern of this gene and comment on the result?

2) Very recently, a ZBTB24 DNA binding motif was identified in human cells (Thompson et al., 2018). Provided that this motif is the same in zebrafish, is it present in sat1 repeats ? And in *Cdca7*, if it happens to be also a *Zbtb24* target in fishes?

3) DNA methylation analyses by ERRBS: using this technique, the authors conclude that *Zbtb24* mostly targets pericentromeric repeats in the zebrafish genome. However, this method provides a very low coverage for regions that are not CpG-rich. ICF type 2 patients (mutated for the ZBTB24 gene) were recently shown to have DNA methylation defects outside of pericentromeric repeats, at CpG-poor regions with heterochromatic features. The authors should acknowledge the limitation of their method in capturing this type of genomic sequences.

4) DNA methylation analyses of sat1 repeats in *Zbtb24* KO: several somatic tissues were analyzed (brain, muscle and skin). What about germ cells? Sperm DNA methylation should be analyzed (see point 4 in this regards). Protocols have been previously published about zebrafish sperm DNA extraction.

5) Phenotype: the authors report in the Discussion that they could not address as to whether the maternal store of *Zbtb24* could mask requirement in DNA methylation establishment because mutant zebrafishes were sterile (I guess this concerns the 10% survivors). Both females *and* males? Regarding the tight link between DNA methylation and fertility in other models, more information about this sterility phenotype is required: are mature germ cells produced (ovary and testis histological sections could give an quick answer)? Of course if males lack sperm, then the analysis requested in #2 becomes invalid.

6) The ability of sat1 repeat transcripts to trigger the expression of ISG genes was assessed with a mixture of sense and antisense RNAs (Figure 5F). The authors should also inject separately sense and antisense products to assess their relative effect on the interferon response and gain mechanistic insights into the origin of this response.

7) One point I find missing from the Discussion: what do we know about the dynamics of DNA methylation during early zebrafish development? Is there something specific going on at two weeks, when *Zbtb24* function becomes essential? See data from the labs of B. Cairns and R. Lister.

Minor Comments:

- Msp1 should be spelled MspI.

- Age at which the ERRBS was performed: the text says at 25 wpf (*Zbtb24* mutants exhibit modest reductions in 5mC at non-pericentromeric sequences), Materials and methods reports 24 wpf.

- DMR call: information is missing about the criteria: size? Sliding window?

*Reviewer #2:*

Vertebrate genomes display high levels of DNA methylation covering repetitive elements, inactive cis-regulatory sequences, intergenic regions, and gene bodies. Numerous functional studies have demonstrated the importance of proper DNA methylation patterning for diverse developmental processes. Nevertheless, such studies frequently suffer from the inability to determine which loci in particular were causal for the phenotypes observed upon the loss of activity of proteins that deposit, read, and write DNA methylation patterns.

The study by Rajshekar et al. provides an exciting system for exploring how loss of DNA methylation from pericentromeric satellite repeats affects vertebrate development. Here the authors mutated the zebrafish *zbtb24* gene, which was previously implicated in Immunodeficiency, Centromere and Facial abnormalities (ICF) syndrome in humans. Importantly, mammalian models for this disorder are either embryonically lethal, or they exhibit immediate post-natal lethality. Using a combination of elegant functional and genome-wide approaches the authors demonstrate how pericentromeric hypomethylation elicits an interferon response in the affected animals. Overall, the presented work is well executed and the manuscript is clearly and concisely written. Significant efforts have been placed in the preparation of the figures, which will make the results easy to understand even for a non-specialist reader. My suggestions for improvement are relatively minor:

1) It would be useful to discuss in more detail what the proposed function of ZBTB24 is. Is it a DNA methylation reader, a chromatin remodeller (I note that it has a helicase domain)? Also, a couple of sentences describing the potential reasons for differences in the observed zebrafish and mouse phenotypes upon the loss of ZBTB24, would be welcome.

2) The ERRBS data are of high interest and will likely be reused in similar studies. I would urge the authors to include more details as to how these libraries were generated and mapped (i.e. how many PCR cycles, which BS-conversion kit, and mapping/mC calling parameters).

3) The scatter plot in Figure 2E leaves the impression as if the majority of CpGs are reduced in the mutant for some 10-20%. This is, of course, difficult to determine by just looking at the scatterplot and even more so if no legend depicting the range of densities is included in the figure. I feel that RRBS data would require some further analysis to unequivocally demonstrate the proposed small magnitude of the changes between the wt and the mutant. I would suggest representing the distributions of 5mC levels at single CpGs in wt and mutant triplicates with boxplots, and stacked bar plots (i.e.% of CpGs with low: 0-0.2, medium: 0.2-0.8, and high: > 0.8 5mC levels). These, and perhaps other analysis, will be better suited for demonstrating the magnitude of changes in 5mC between the wt and the mutant. Also, could the authors comment on the proposed mechanism for this apparent genome-wide 5mC loss in the mutant?

4) Is this global 5mC loss correlated with any genomic feature in particular?

5) I would remove the non-significant P values from Figures 3, 4 and 5 and just state "n.s" instead. This will allow the reader to immediately focus on the significant changes. I would consider replacing all the P values with *, **, or *** – denoting increasing significance levels, and just state the P value ranges in figure legends.

Overall, this is a strong paper on a topic of high interest that sets the stage for exciting future studies. For example, it will be interesting to see how specific the *mda5* response is and what other types of RNA can activate these pathways.

*Reviewer #3:*

The manuscript describes a model of ICF syndrome in zebrafish by mutating the ZBTB24 gene, previously described as an ICF-gene. This model recapitulates an ICF hallmark that is pericentromeric (SAT1) hypomethylation. Authors describe that hypomethylation does not affect other genomic regions. As a consequence SAT1 transcripts are detected, parallel to expression of ISGs (IFN response) that may be related to autoimmunity. They show that the IFN pathway dsRNA sensor MDA5-MAVS is required for those effects, sensing SAT1 transcripts.

It is not clear how ZBTB24 mutant can generate specific demethylation of pericentromeric repeats only, nor what is the relation between ISG upregulation and ICF etiology. Nonetheless, experiments are consistent and proof well the relation between hypomethylation, SAT1 expression and IFN pathway activation through the dsRNA sensors. Similar experiments have been performed elsewhere with DNA methylation inhibitor drugs in human cancer cells (or by histone H1 depletion).

Major comments:

1) It would be nice to test whether aza-dC causes the same effects as ZBTB24 mutation in zebrafish.

2) In the experiment of SAT1 dsRNA injection, it might be done in parallel with sense and antisense RNA separately to proof dsRNA is required.

3) It would have been nice to analyze by ChIP what changes occur in SAT1 upon ZBTB24 mutation apart from hypomethylation, for example in terms of repressive histone marks.

Minor comments:

Figure 6A and C might be A and B (3 and 6 weeks, respectively). RNAseq on C.

---

## [Author Response]

Reviewer #1:

[…] The manuscript is clear and concise, and the conclusions well supported by the data. The genetic approaches in particular are very solid.I have a few requests for the authors though, to improve the clarity and breadth of the findings.1) It has been shown in mammals that Zbtb24 controls the expression of Cdca7, whose mutations are also involved in the ICF syndrome. Is it also the case in zebrafish? Could the authors look in their RNA seq data from Zbtb24 KO the expression pattern of this gene and comment on the result?

Our RNA-seq data does indicate that *cdca7* expression is downregulated in *zbtb24^Δ/Δ^* mutant zebrafish and we have confirmed this result by qRT-PCR. The relevant data is now included in Figure 6—figure supplement 1 and we include a discussion of this point in the fifth paragraph of the Discussion.

2) Very recently, a ZBTB24 DNA binding motif was identified in human cells (Thompson et al., 2018). Provided that this motif is the same in zebrafish, is it present in sat1 repeats ? And in Cdca7, if it happens to be also a Zbtb24 target in fishes?

Thompson et al., 2018, identified a *ZBTB24* DNA binding motif: AGGTCCTGGCAG, in human cells. We used Find Individual Motif Occurrences (FIMO) (Grant et al., 2011), to search for occurrence of this motif in the zebrafish genome. We did not find this motif in the promoter or gene body of *cdca7* or in Sat1 repeats. We now include this point in our Discussion (fifth paragraph).

3) DNA methylation analyses by ERRBS: using this technique, the authors conclude that Zbtb24 mostly targets pericentromeric repeats in the zebrafish genome. However, this method provides a very low coverage for regions that are not CpG-rich. ICF type 2 patients (mutated for the ZBTB24 gene) were recently shown to have DNA methylation defects outside of pericentromeric repeats, at CpG-poor regions with heterochromatic features. The authors should acknowledge the limitation of their method in capturing this type of genomic sequences.

We now highlight this limitation in our Discussion (third paragraph).

4) DNA methylation analyses of sat1 repeats in Zbtb24 KO: several somatic tissues were analyzed (brain, muscle and skin). What about germ cells? Sperm DNA methylation should be analyzed (see point 4 in this regards). Protocols have been previously published about zebrafish sperm DNA extraction.

We agree that analysis of germ cells is of interest, and we have performed these experiments. However, we were somewhat hesitant to include this data in the original manuscript as the results are unexpected and raise questions that we feel are outside the scope of the current manuscript. While far less sperm is recovered from *zbtb24* mutants compared to wildtype, some sperm can be recovered. Unexpectedly, we find that methylation levels in the recovered sperm from *zbtb24^Δ/Δ^* males are comparable to wildtype. This is in contrast to all somatic tissues we have examined. We now include this finding in Figure 2—figure supplement 1C-D. A description of the result is presented in the first paragraph of the subsection “Progressive methylation loss at pericentromeric repeats in *zbtb24* mutants”and some discussion is included in the second paragraph of the Discussion, our methods for sperm recovery are provided in the subsection “Sperm Count”.

5) Phenotype: the authors report in the Discussion that they could not address as to whether the maternal store of Zbtb24 could mask requirement in DNA methylation establishment because mutant zebrafishes were sterile (I guess this concerns the 10% survivors). Both females and males? Regarding the tight link between DNA methylation and fertility in other models, more information about this sterility phenotype is required: are mature germ cells produced (ovary and testis histological sections could give an quick answer)? Of course if males lack sperm, then the analysis requested in #2 becomes invalid.

As discussed in response to comment 4 above, the sterility in phenotype *zbtb24^Δ/Δ^* mutants appears complicated, and substantial investigation beyond the scope of this manuscript may be needed to fully understand it. All attempts to cross male and female mutant fish with each other or to outcross male or female mutants to wildtype have been unsuccessful. We do find that the testes in *zbtb24^Δ/Δ^* mutants are thinner compared to wildtype siblings and that the sperm count in *zbtb24^Δ/Δ^* mutants is significantly reduced when we dissect and manually grind up whole testes to extract sperm. However, histological sections reveal overtly normal male and female gonads with formation of what appear to be mature sperm and eggs. For reasons that are unclear to us, we have been unable to recover these seemingly mature germ cells using traditional methods for in vitro fertilization (which we routinely employ in the lab). We now provide this characterization as *zbtb24*supplementary material (Figure 1—figure supplement 3) and describe these findings in the subsection “Mutation of zebrafish causes ICF syndrome-like phenotypes”. Methods for histology and sperm counting are included in the subsections “Histology” and “Sperm count” respectively.

6) The ability of sat1 repeat transcripts to trigger the expression of ISG genes was assessed with a mixture of sense and antisense RNAs (Figure 5F). The authors should also inject separately sense and antisense products to assess their relative effect on the interferon response and gain mechanistic insights into the origin of this response.

We have now included additional experimental data for sense and antisense transcripts injected individually. In contrast to combined injection of sense and antisense RNA which reproducibly yielded significant upregulation of ISGs, we find lower level and more variable induction of only some ISGs when either sense or antisense RNA is injected alone. We include this data in Figure 5—figure supplement 2 and we include potential interpretations of this result in the (subsection **“**Pericentromeric RNA transcripts are sufficient to trigger the interferon response in *zbtb24* mutants”, last paragraph and Discussion, seventh paragraph).

7) One point I find missing from the Discussion: what do we know about the dynamics of DNA methylation during early zebrafish development? Is there something specific going on at two weeks, when Zbtb24 function becomes essential? See data from the labs of B. Cairns and R. Lister.

Dynamics of DNA methylation across early zebrafish development have been extensively characterized (Jiang et al., 2013; Potok et al., 2013, Bogdanovic, 2016). But these analyses have focused on developmental stages prior to 48 hpf. To the best of our knowledge genome-wide methylome analyses on zebrafish has not been performed at 2 weeks post fertilization or in the window surrounding this time point. As such, we are unaware of any developmental or methylation milestones that can account for the onset of hypomethylation in *zbtb24* mutants. Rather, we speculate that the onset of methylation loss at this stage partly reflects the need to deplete maternally loaded *zbtb24* prior to unmasking of the *zbtb24* mutant phenotype and partly reflects the culmination of minor methylation losses due to lower fidelity maintenance over many rounds of cell division. We now include text to this effect in the fourth paragraph of the Discussion.

Minor Comments:- Msp1 should be spelled MspI.

Msp1 has been changed to MspI in the manuscript.

- Age at which the ERRBS was performed: the text says at 25 wpf (Zbtb24 mutants exhibit modest reductions in 5mC at non-pericentromeric sequences), Materials and methods reports 24 wpf.

We apologize for this oversight on our part. We have revised all references to ERRBS data to indicate that animals were 6 months old.

- DMR call: information is missing about the criteria: size? Sliding window?

Additional details of DMR calling, as well as a reference which includes extensive details of our approach are now included in the subsection “Enhanced Reduced Representation Bisulfite Sequencing (ERRBS)”.

Reviewer #2:

[…] Overall, the presented work is well executed and the manuscript is clearly and concisely written. Significant efforts have been placed in the preparation of the figures, which will make the results easy to understand even for a non-specialist reader. My suggestions for improvement are relatively minor:1) It would be useful to discuss in more detail what the proposed function of ZBTB24 is. Is it a DNA methylation reader, a chromatin remodeller (I note that it has a helicase domain)? Also, a couple of sentences describing the potential reasons for differences in the observed zebrafish and mouse phenotypes upon the loss of ZBTB24, would be welcome.

We now discuss the possible roles of ZBTB24 in regulating DNA methylation in the fifth paragraph of the Discussion.We really don’t understand why mouse and zebrafish *zbtb24* mutant phenotypes are so different and prefer not to speculate.

2) The ERRBS data are of high interest and will likely be reused in similar studies. I would urge the authors to include more details as to how these libraries were generated and mapped (i.e. how many PCR cycles, which BS-conversion kit, and mapping/mC calling parameters).

We apologize for the omission and now include the additional information on the methods including PCR cycles, BS- conversion kit and mapping/mC calling parameters in the subsection “Enhanced Reduced Representation Bisulfite Sequencing (ERRBS)”.

3) The scatter plot in Figure 2E leaves the impression as if the majority of CpGs are reduced in the mutant for some 10-20%. This is, of course, difficult to determine by just looking at the scatterplot and even more so if no legend depicting the range of densities is included in the figure. I feel that RRBS data would require some further analysis to unequivocally demonstrate the proposed small magnitude of the changes between the wt and the mutant. I would suggest representing the distributions of 5mC levels at single CpGs in wt and mutant triplicates with boxplots, and stacked bar plots (i.e.% of CpGs with low: 0-0.2, medium: 0.2-0.8, and high: > 0.8 5mC levels). These, and perhaps other analysis, will be better suited for demonstrating the magnitude of changes in 5mC between the wt and the mutant. Also, could the authors comment on the proposed mechanism for this apparent genome-wide 5mC loss in the mutant?

We now also present% methylation as violin plots for individual *zbtb24* WT and mutant samples (Figure 3F). In the third paragraph of the Discussion, we now include text suggesting that low level methylation changes outside of the pericentromeres may reflect more modest requirements for *zbtb24* in maintaining methylation at these sequences.

4) Is this global 5mC loss correlated with any genomic feature in particular?

We now include violin plots comparing methylation for different classes of genomic features including CpG islands, promoters and exons in Figure 2—figure supplement 4 and discuss this data in the subsection “*Zbtb24* mutants exhibit modest reductions in 5mC at non-pericentromeric sequences”.

5) I would remove the non-significant P values from Figures 3, 4 and 5 and just state "n.s" instead. This will allow the reader to immediately focus on the significant changes. I would consider replacing all the P values with *, **, or *** – denoting increasing significance levels, and just state the P value ranges in figure legends.

We feel that providing exact p values promotes the most transparency.

Overall, this is a strong paper on a topic of high interest that sets the stage for exciting future studies. For example, it will be interesting to see how specific the mda5 response is and what other types of RNA can activate these pathways.

Reviewer #3:

[…] It is not clear how ZBTB24 mutant can generate specific demethylation of pericentromeric repeats only, nor what is the relation between ISG upregulation and ICF etiology. Nonetheless, experiments are consistent and proof well the relation between hypomethylation, SAT1 expression and IFN pathway activation through the dsRNA sensors. Similar experiments have been performed elsewhere with DNA methylation inhibitor drugs in human cancer cells (or by histone H1 depletion).Major comments:1) It would be nice to test whether aza-dC causes the same effects as ZBTB24 mutation in zebrafish.

As requested, we exposed zebrafish embryos to 5 azaC and observed upregulation of interferon stimulated genes. This experiment is presented in Figure 3—figure supplement 2 and discussed in the subsection “Mutation of *zbtb24* causes activation of innate immune response genes”. Methods are provided inthe subsection **“**5-aza-dC Treatment”. This is finding consistent with earlier reports in the literature demonstrating activation of an innate immune response in *dnmt1^-/-^* zebrafish embryos undergoing global DNA hypo methylation (Chernyavskaya et al., 2017). This paper, along with other findings linking global DNA hypomethylation to innate immunity are referenced in the first paragraph of the Introduction.

2) In the experiment of SAT1 dsRNA injection, it might be done in parallel with sense and antisense RNA separately to proof dsRNA is required.

See response to reviewer 1 comment 6.

3) It would have been nice to analyze by ChIP what changes occur in SAT1 upon ZBTB24 mutation apart from hypomethylation, for example in terms of repressive histone marks.

We now include ChIP-qPCR and immunofluorescence data assessing H3K9me3 and H3K27me3 in Figure 2—figure supplement 2.Methods are presented inthe subsection “Chromatin Immunoprecipitation (ChIP)” and we highlight these findings in the last paragraph of the subsection “Progressive methylation loss at pericentromeric repeats in *zbtb24* mutants”. We did not observe any obvious changes in these repressive histone marks.

Minor comments:Figure 6A and C might be A and B (3 and 6 weeks, respectively). RNAseq on C.

We have made this change.